# Combining the Effects of Global Warming, Land Use Change and Dispersal Limitations to Predict the Future Distributions of East Asian Cerris Oaks (*Quercus* Section *Cerris*, Fagaceae) in China

Yuheng Chen [ID], Yao Li [ID] and Lingfeng Mao *

Co-Innovation Center for Sustainable Forestry in Southern China, College of Biology and the Environment, Nanjing Forestry University, Nanjing 210037, China; edenchen1226@outlook.com (Y.C.); liyaolisantu@njfu.edu.cn (Y.L.)
* Correspondence: maolingfeng2008@163.com

**Abstract:** Species shift their ranges in response to climate change (CC). However, they may not be able to track optimal conditions as soon as possible, due to limited dispersal ability or habitat fragmentation, caused by land use and land cover change (LULC). This study aimed to explore the combined impacts of CC, LULC and dispersal limitations on the future range dynamics of *Quercus acutissima* Carruth., *Q. variabilis* Blume and *Q. chenii* Nakai, three dominant Cerris oak tree species in warm-temperate and subtropical deciduous forests of China. We used the Maximum Entropy (Maxent) algorithm to predict the suitable habitats for the years 2050 and 2070, under three representative concentration pathways (RCPs). Habitat fragmentation patterns were examined to assess the influence of LULC. Two migration scenarios (full- and partial-migration) were compared to evaluate the effect of dispersal limitations. We found that annual precipitation (AP), minimum temperature in the coldest month (MTCM) and temperature seasonality (TS) play a key role in determining the present distributions of *Q. chenii*, while AP, MTCM and annual mean temperature (AMT) contribute the most to the distribution models of *Q. variabilis* and *Q. acutissima*. For all the three species, LULC will increase the level of habitat fragmentation and lead to the loss of core areas, while limited dispersal ability will restrict the accessibility of future potentially suitable habitats. Under the scenarios of CC and LULC, the suitable areas of *Q. chenii* will decrease sharply, while those of *Q. variabilis* in South China will become unsuitable. Our findings highlight the importance of considering dispersal ability, as well as land use and land cover change, for modeling species' range shifts in the face of global warming. Our study also provides vital information for guiding the management of East Asian Cerris oaks in China; *Q. chenii* should be listed as a species requiring priority protection, and the threatened habitats of *Q. variabilis* should be protected to buffer the impacts of CC and LULC.

**Keywords:** climate change; dispersal limitation; land use and land cover change; Maxent; *Quercus*; species distribution model

## 1. Introduction

Global climate change (CC) has become one of the major threats to biodiversity and conservation [1,2]. CC caused by human activities, especially greenhouse gas emissions, results in alterations in temperature and precipitation patterns, affecting plant growth and development across the entire life cycle [3,4]. On one hand, species may respond to CC by shifting their ecological niches through plastic changes to avoid range contractions and extinctions [5,6]. On the other hand, many species' ranges have shifted to prevent being affected by adverse climatic conditions [7].

Given that climate has been documented to play a key role in determining large-scale species distributions [8,9], ecologists often use species distribution models (SDMs) to

quantify the relationships between existing occurrence records and climatic factors through multivariate algorithms [5]. These models can generate a prediction for species' future habitat suitability, thus, providing us with a useful tool for evaluating habitat vulnerability in the face of global warming [10,11]. However, as most SDMs are targeted at potential ecological niches, the majority of SDM studies prefer to apply full-migration scenarios in predicting species' range shifts in response to CC [5,12]. Indeed, species may not be able to track favorable habitats as soon as possible due to dispersal limitations. Thus, the existing models may hinder our accurate assessment of the real risk to a species [13]. Recently, more researchers have noticed this issue and tried to use various modeling approaches to incorporate migration scenarios into SDMs, for example, future projections for fir species in Southwest China and amphibians in the Himalayas [14,15]. These predictions took species' dispersal abilities into account and improved the accuracy of SDMs, which will guide the conservation of endangered species more reliably than the models only considering full-migration or no-migration scenarios.

The migration of a species is not only determined by the species' dispersal ability, but also by the levels of habitat fragmentation and connectivity [16]. Land use and land cover change (LULC) reflects the changes in land cover [17]. Different land types have different species composition and vegetation cover, and constantly transform local microclimates [18,19]. Hence, although many areas are climatically suitable for species' survival, effective dispersal may be severely restricted due to strong geographical isolation caused by LULC [20]. With the increase in human activities, 10% to 20% of natural grasslands and forests are expected to be replaced by agricultural and urban infrastructure by 2050 [21], which will accelerate habitat loss and fragmentation for most wild species [22,23]. In this sense, it is essential to explore the combined effects of LULC and dispersal limitations on the changes of species' ranges in response to CC.

Oaks (*Quercus* spp.) are one of the most common broad-leaved tree species in the Northern Hemisphere, usually occupying a (co-)dominant position in local forest ecosystems [24]. This genus has recently been divided into two subgenera and eight sections [25], of which sect. *Cerris* is mainly distributed in Eurasia and comprises 13 species. Among these species, only *Quercus acutissima* Carruth., *Q. variabilis* Blume and *Q. chenii* Nakai are native to East Asia, while the other 10 species are endemic to western Eurasia [25]. In this study, we chose to predict the range shifts of the three East Asian Cerris oak species because they are among the dominants of East Asian warm-temperate and subtropical deciduous forests. Previous studies have shown that historical climate change has significantly affected the distributions of these oak forests. Global cooling during the Last Glacial Maximum (LGM, ~22,000 years ago) may have resulted in the southward retreat of temperate deciduous oak forests to between 22° N and 31° N. In contrast, climate warming during the mid-Holocene (~6000 years ago) may have caused their northward expansion [26,27].

Currently, *Q. acutissima* and *Q. variabilis* are widely distributed across East Asia, while *Q. chenii* exhibits a narrow distribution in eastern subtropical China [26,27]. The areas of pure stands of *Q. variabilis* and *Q. acutissima* in China are estimated to be 13,634.6 km$^2$ and 7451.1 km$^2$, respectively, while the forested area with *Q. variabilis*, *Q. acutissima* or *Q. chenii* as a (co-)dominant species is more than 70,000 km$^2$ [28]. As major contributors to ecosystem function, the three oak species are regarded as indicator species for assessing local forest health [29]; in many cases, the disappearance of oaks in forests means the disappearance of endemic species and a reduction in species diversity [30,31]. Furthermore, East Asian Cerris oaks provide local residents with wood products, food, and fuel, and have been listed as one of the precious wood species by the government [28]. Given that East Asian Cerris oaks are not only ecologically but also economically important [26,27], it is reasonable to use them as a model to understand the impacts of CC, LULC and dispersal limitations on future range dynamics of keystone forest tree species in East Asia.

Here, we integrate both SDMs and migration models to explore the roles of CC, LULC and dispersal ability in shaping the range dynamics of East Asian Cerris oaks. We aimed to (1) determine the key climatic factors that affect habitat changes in response to CC;

(2) explore the combined effects of LULC and dispersal limitations on range shifts of East Asian Cerris oaks; (3) identify habitats potentially threatened by CC and LULC and provide guidance for the future management of East Asian Cerris oak forests.

## 2. Materials and Methods

### 2.1. Species Occurrence Records and Climatic Data

The study was conducted in China, which is in the eastern part of Asia. Species occurrence records of the three East Asian Cerris oaks were obtained from the Chinese Virtual Herbarium (CVH) [32], Plant Photo Bank of China (PPBC, [33]) and field investigation. We did not use the data of Global Biodiversity Information Facility (GBIF) because they are a subset of the CVH database. For occurrences lacking geographic coordinates, we used the Getpoint tool of Baidu Maps to complement the latitude and longitude information according to explicit geographic locations [34]. Moreover, any duplicate records or those of introduction and cultivation were excluded. Because SDMs require input data to be spatially independent so that they can perform well, we cleaned our data to ensure that only one occurrence record per species was used within each grid cell at a resolution of 0.05° [35]. Finally, we obtained 402, 52, and 431 records for *Q. acutissima*, *Q. chenii*, and *Q. variabilis*, respectively (Table S1). Maps of these *Quercus* species occurrences were visualized in ArcGIS 10.3 (Figure 1) [36].

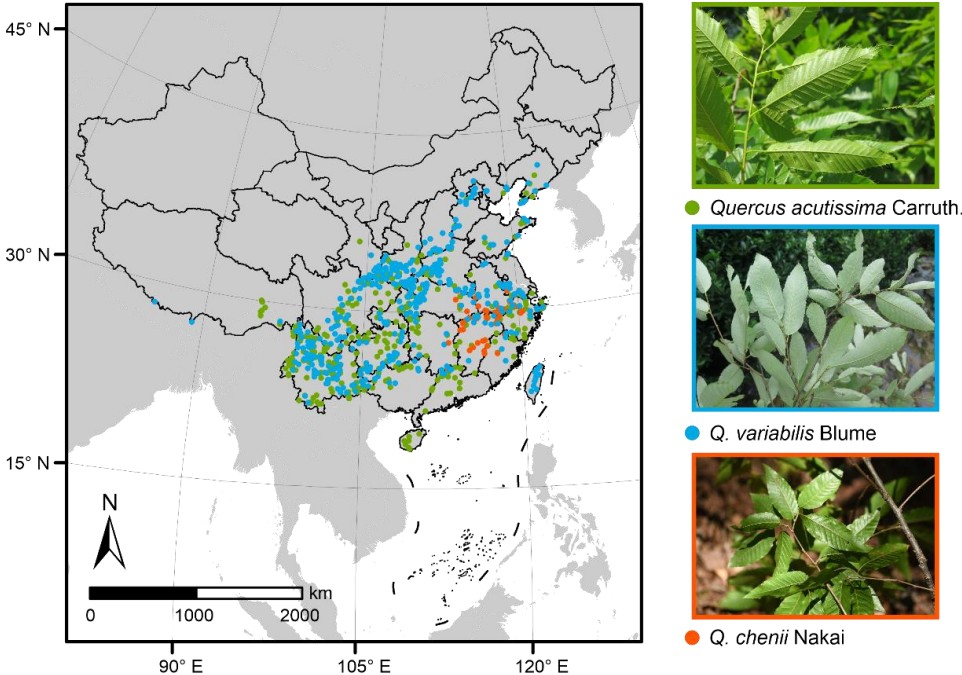

**Figure 1.** Maps of occurrence records and leaf photos (taken by Y.L.) of the three East Asian Cerris oaks. The green, blue and red dots represent the presence points of *Quercus acutissima* Carruth. (*n* = 402), *Q. variabilis* Blume (*n* = 431) and *Q. chenii* Nakai (*n* = 52), respectively.

The raster layers of 19 bioclimatic predictors (1970–2000) were obtained from the WorldClim version 2.1 database at a spatial resolution of 2.5 min and then projected to the GCS_WGS_1984 system (see Table S2 for specific factors) [37]. Highly collinear variables allow alternative model structures to yield very similar model fits [38]. To avoid the interference of multicollinearity between variables [39], we examined the cross-correlation of the 19 variables using the 'band collection statistics' function in ArcGIS and eliminated the highly correlated (|Pearson *r*| ≥ 0.8) climatic variables. Although the correlation between annual precipitation (AP) and precipitation of driest quarter (PDQ) exceeded 0.8 (|Pearson *r*| = 0.859), we retained these two variables to investigate the impact of extreme precipitation on species distributions. Finally, out of the total 19 variables,

only seven variables were selected as predictors (Table S3), including AP, PDQ, annual mean temperature (AMT), isothermality (IT), temperature seasonality (TS), minimum temperature of coldest month (MTCM) and precipitation seasonality (PS).

## 2.2. SDMs Incorporating CC

We used a Maximum Entropy approach as implemented in Maxent version 3.4.1 [40] to simulate the modern distributions of East Asian Cerris oaks in China. Maxent can predict probability distribution of a given occurrence dataset based on presence-only points and predictors [41]. We chose the Maxent method because it is one of the best-performing models among all the SDMs [42]. To reduce uncertainty caused by sampling artefacts, we randomly divided occurrence data into training data (75%) and validation data (25%). We ran the model 20 times independently and used the subsampling method to validate the robustness of the models. The default setting of Maxent used automatic method to generate feature types, 5000 as maximum number of iterations, and 10,000 as maximum number of background points for seeking the optimal solution.

We projected the model to two future periods (2050 and 2070) to predict the impacts of CC on range shifts of East Asian Cerris oaks. Three representative concentration pathways (RCPs) were considered, including RCP 2.6, RCP 4.5, and RCP 6.0, representing different concentration trajectories from eco-friendly (RCP 2.6) to bad case (RCP 6.0) [43]. In the RCP 2.6 scenario, mean annual temperature in China is estimated to increase by 1.36 °C from current to 2070, while in the RCP 4.5 and RCP 6.0 scenarios, it is predicted to increase by 2.13 °C and 2.36 °C, respectively. These data were generated using the global climate model (GCM) BCC-CSM1-1 (Beijing Climate Centre, China Meteorological Administration), which is considered one of the more suitable GCMs for climate change research in China [44]. All the layers for future climatic variables were downloaded from WorldClim version 2.1 database at a spatial resolution of 2.5 min [37]. Finally, we used the resampling function based on the nearest neighbor method in ArcGIS 10.3 to convert the resolution of the climate data to 0.05°, matching with the resolution of land use and land cover data (see below).

We used the maximum training sensitivity plus specificity (MTSS) threshold to convert the continuous suitability scores (range of 0 to 1) of the Maxent outputs into binary suitability: unsuitable habitats (<MTSS), and suitable habitats (≥MTSS) [45]. We used the variable importance index provided by Maxent to evaluate the contributions of each variable to the model. For each environmental variable in turn, the values of the variable on training presence and background data were randomly permuted and the model was reevaluated using the permuted data and then normalized the AUC loss to percentages [41]. We adopted the area under the receiver operation curve (AUC) to verify the accuracy of each model. An AUC value greater than 0.9 often indicates that the model fits the observed dataset well [5].

## 2.3. SDMs Incorporating LULC

To predict the impacts of LULC on range dynamics of East Asian Cerris oaks, we extracted the Chinese land use and land cover data for now, 2050 and 2070, from an open-source global future land use dataset [17]. This dataset is obtained based on a global change analysis model (GCAM) and a land use spatial downscaling model (Demeter version 1.0.0) under the framework of the Shared Socioeconomic Pathways (SSPs) [46]. The grid shows all different land use types in the form of proportion. Compared with the existing similar datasets, this dataset has a higher spatial resolution (0.05° × 0.05°) and considers uncertainties from the forcing climates [17]. To investigate the habitat changes of East Asian Cerris oaks in China, we extracted the land use categories 'coniferous forest' (CF) and 'broad-leaved forest' (BF) in the dataset, which represent the proportion changes of global coniferous forests and broad-leaved forests in each grid in the future. To be comparable with the three CC scenarios, we grouped SSPs 2 with the RCP 2.6 scenario, SSPs 3 with the RCP 4.5 scenario, and SSPs 4 with the RCP 6.0 scenario.

To assess the impacts of LULC on habitat change, we calculated the number of patches (NP) and core area (CA, $\times 10^4$ km$^2$) of habitats through the 'SDMtools' package in R

4.0.2 [47]. NP reflects the level of habitat fragmentation, and the increase in NP indicates that the original habitat is transforming to many isolated fragments [48]. CA is the area where its four parallel neighbors are not disturbed by the unsuitable habitats. The increase in CA means more undisturbed habitats with high population connectivity [49]. We first compared the differences in NP and CA between SDMs considering CC and those considering both CC and LULC (i.e., CC + LULC), and then we used the CC + LULC model to predict the changes of future habitats for East Asian Cerris oaks in China.

### 2.4. SDMs Incorporating Dispersal Ability

To simulate the effects of dispersal ability on the accessibility of future suitable habitat areas, the following two different migration rates were assigned to all species: full migration (FM, unlimited m/year) without any dispersal limitations, and partial migration (PM, 500 m/year) based on literature [50]. The FM model was obtained directly from the Maxent outputs of the CC + LULC model applying a species-specific MTSS threshold, coinciding with the most optimistic assumption that species could colonize all suitable habitats under CC and LULC [45].

For the PM scenario, we used the KISSMig model in R 4.0.2 [13], a simple $3 \times 3$ raster-based stochastic approach to simulate dynamic changes in species distributions on top of habitat suitability maps generated by the CC + LULC model. Under this model, species disperse based on the current range at a given ratio of migration. The algorithm is generally used to detect the overall migration pattern through time and provides rough estimates of the general migration rates [13]. It is widely used in niche modeling of species with limited vagility (e.g., endangered species and plants) [14,51]. Alpha-shapes associated with the original presence points were used to restrict the current species range [15]. We used binary suitability values (unsuitable: 0, suitable: 1) for migration simulations instead of using quantitative suitability values [14]. From the present (2020) to the years 2050 and 2070, the migration distance was set to 15 km and 25 km, respectively.

## 3. Results

### 3.1. Model Performance and Key Climatic Factors

The mean AUC values ($\pm$SD) on the observed dataset of each species were higher than 0.9 (*Q. acutissima*: $0.974 \pm 0.006$; *Q. chenii*: $0.977 \pm 0.004$; *Q. variabilis*: $0.925 \pm 0.004$), indicating that our SDMs had an excellent overall prediction ability. The SDMs only considering climatic variables showed that AP, MTCM and AMT were the most important variables for predicting the distributions of the two widespread species, *Q. variabilis* and *Q. acutissima* (Figure 2c). Their cumulative relative importance exceeded 75%. For the narrowly distributed species *Q. chenii*, AP, MTCM and TS were the most important variables, accounting for 84.6% of the cumulative relative importance, while AMP showed little contribution to predicting the suitable habitats of *Q. chenii* (Figure 2c).

### 3.2. Sensitivity to LULC in SDMs

SDMs showed that the overall range of current habitats did not change significantly after taking the LULC data into account (Figure 3). The importance of the two LULC variables was much lower than that of climatic variables, as indicated by the fact that CF and BF only accounted for 0.4% of the cumulative relative importance in the model of *Q. chenii*, and for less than 10% in models of *Q. acutissima* and *Q. variabilis*. However, after adding the LULC data, the landscape patterns of habitats changed significantly. For all the three species, the values of NP were found to be higher under the CC + LULC scenario than under the CC scenario (Figure 2a), while the values of CA showed an opposite tendency (Figure 2b). The increase in NP was greater in widely distributed species *Q. acutissima* and *Q. variabilis* than in narrowly distributed species *Q. chenii*. The proportion of reduction in CA was also much smaller in *Q. acutissima* and *Q. variabilis* than in *Q. chenii*.

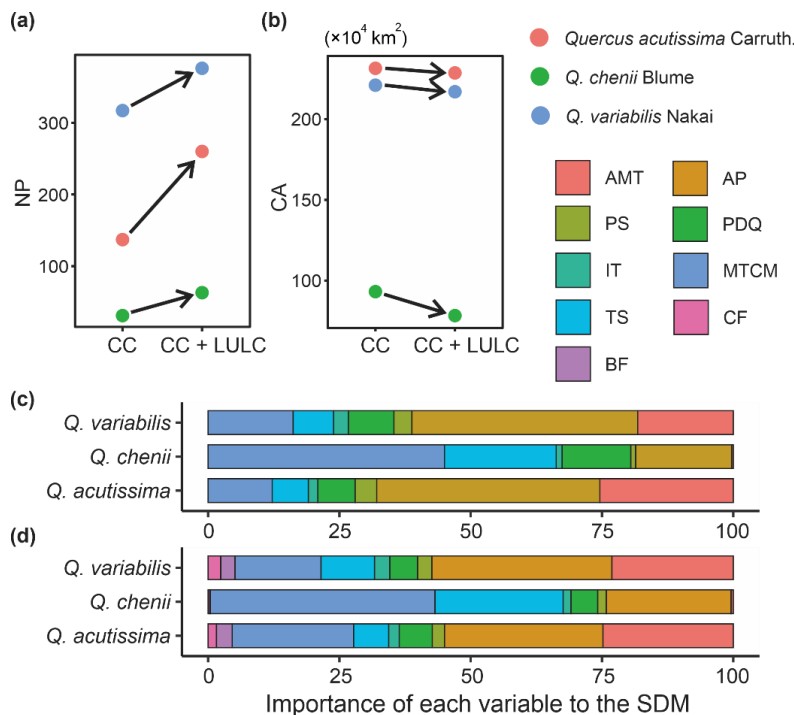

**Figure 2.** (**a**,**b**) Changes of landscape patterns, i.e., number of patches (NP) (**a**) and core area (CA) (**b**), between species distribution models (SDMs) only considering climate change (i.e., the CC model) and those considering CC as well as land use and land cover (LULC) change (i.e., the CC + LULC model). (**c**,**d**) Variable importance index provided by Maxent to evaluate the contributions of each variable to SDMs. (**c**,**d**) show the importance of each variable in the CC model and CC + LULC model, respectively. AMT, annual mean temperature; AP, annual precipitation; BF, broad-leaved forest; CF, coniferous forest; IT, isothermality; MTCM, minimum temperature of coldest month; PDQ, precipitation of driest quarter; PS, precipitation seasonality; TS, temperature seasonality.

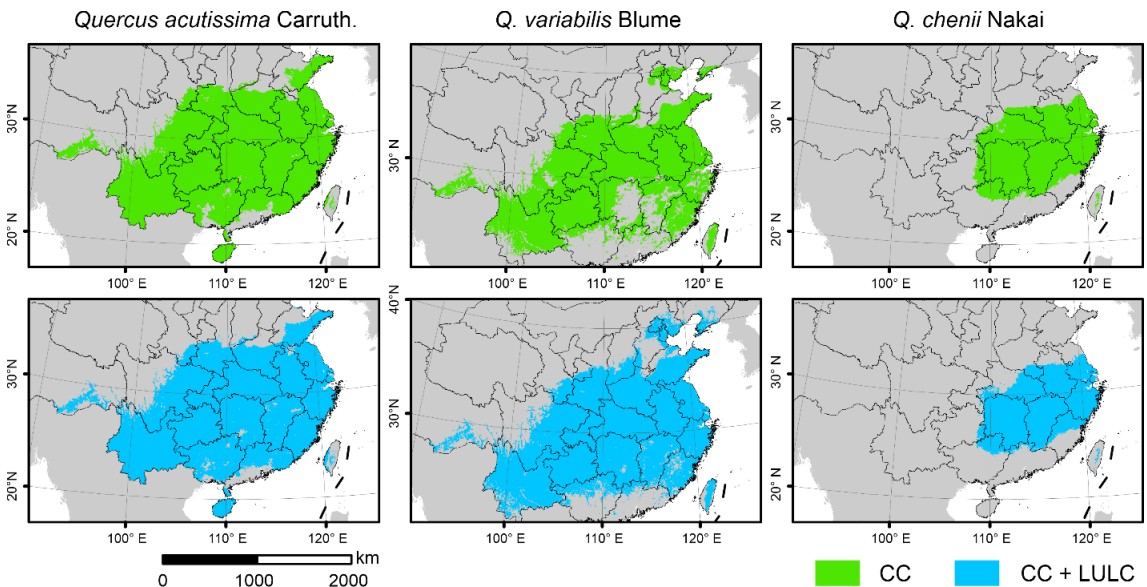

**Figure 3.** Current suitable habitats of the three East Asian Cerris oaks in China predicted by species distributions models (SDMs) only considering climate change (i.e., the CC model, green) and those considering CC as well as land use and land cover (LULC) change (i.e., the CC + LULC model, blue). The species-specific maximum training sensitivity plus specificity (MTSS) thresholds were used to classify the suitable habitats (>MTSS) and unsuitable habitats (<MTSS).

### 3.3. Projected Future Changes in Species Habitats

Under the FM model, both suitable habitat areas and CAs were predicted to change significantly from 2050 to 2070 (Figures 4 and 5, Table 1). For *Q. acutissima*, the area of current suitable habitats was estimated to be $253.33 \times 10^4$ km$^2$. It will increase under all the three CC scenarios (RCP 2.6, RCP4. 5 and RCP 6.0) of 2050, but will decrease to ~$214.33 \times 10^4$ km$^2$ under the RCP 6.0 scenario of 2070. The value of CA will decrease in the future; the loss of CA is predicted to be smaller in the RCP 2.6 scenario than in the other two RCPs.

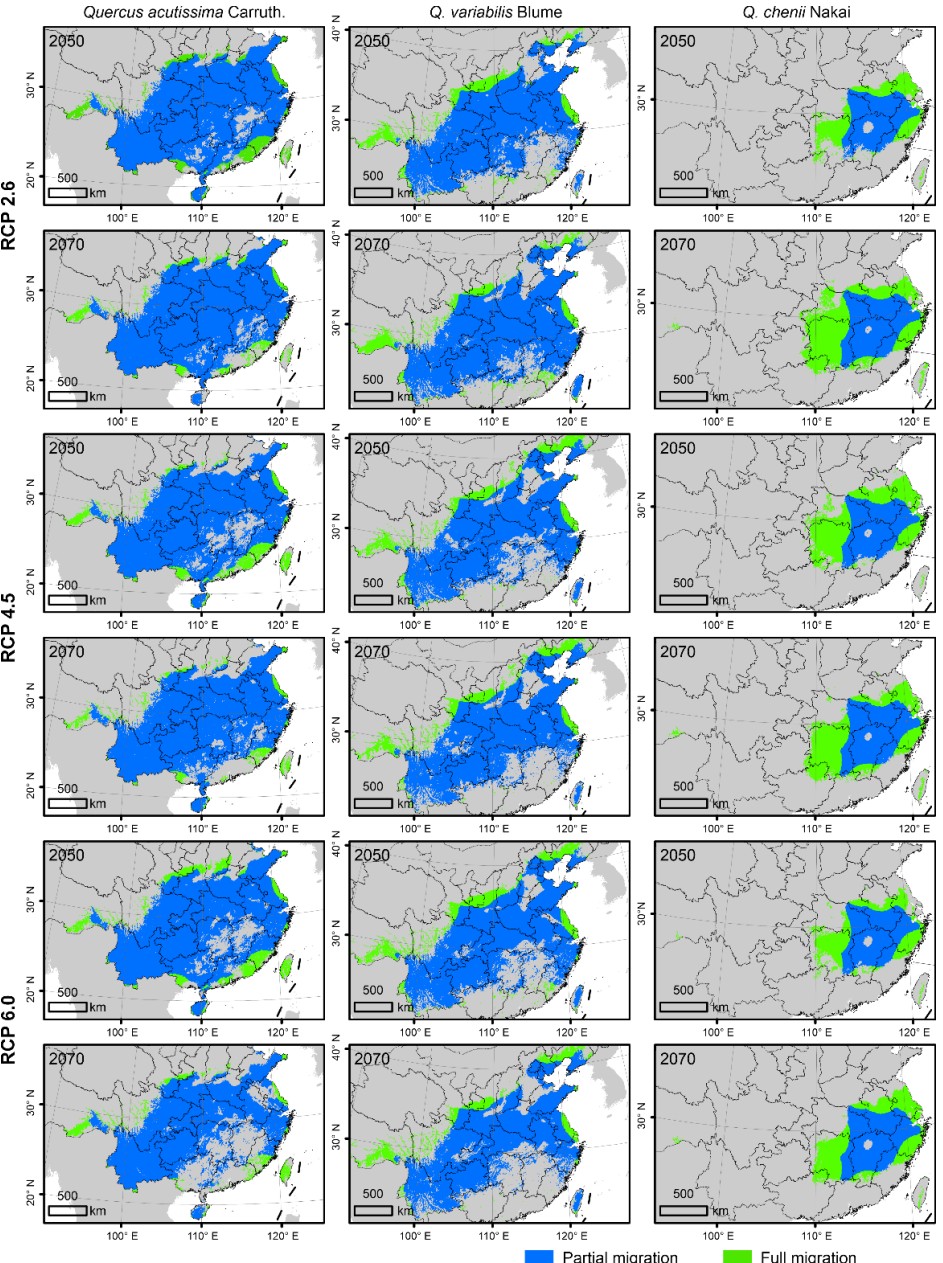

**Figure 4.** Future suitable habitats of the three East Asian Cerris oaks in China in the years 2050 and 2070 under three representative concentration pathways (RCPs), i.e., RCP 2.6, RCP 4.5, and RCP 6.0. They were predicted by species distributions models (SDMs) considering climate change (CC), land use and land cover change (LULC) and dispersal limitations. The green and blue areas represent the suitable habitats under the full migration (FM, i.e., migration unlimited) and partial migration (PM, 500 m/year) scenarios, respectively. The species-specific maximum training sensitivity plus specificity (MTSS) thresholds were used to classify the suitable habitats (>MTSS) and unsuitable habitats (<MTSS).

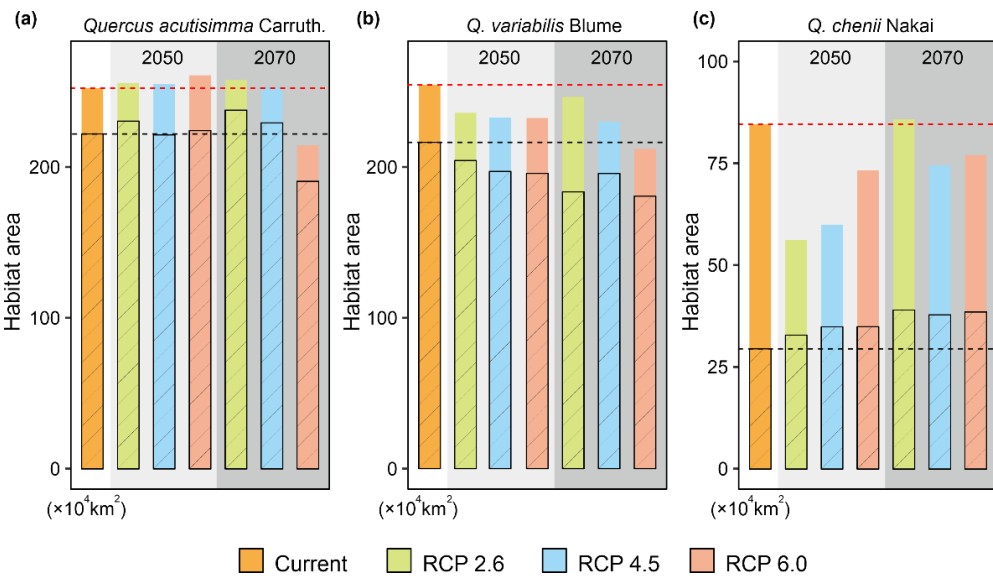

**Figure 5.** Changes of suitable habitat areas for *Quercus acutissima* Carruth. (**a**), *Q. variabilis* Blume (**b**), and *Q. chenii* Nakai (**c**) in China from the present to the years 2050 and 2070 under three representative concentration pathways (RCPs), i.e., RCP 2.6, RCP 4.5, and RCP 6.0. The habitat areas were estimated based on the results of species distributions models (SDMs) considering climate change (CC), land use and land cover change (LULC) and dispersal limitations. The solid and open histograms represent the suitable habitat areas under the full migration (FM, i.e., migration unlimited) and partial migration (PM, 500 m/year) scenarios, respectively. Red and black dashed lines represent the present suitable habitat areas under FM and PM scenarios, respectively. The species-specific maximum training sensitivity plus specificity (MTSS) thresholds were used to classify the suitable habitats (>MTSS) and unsuitable habitats (<MTSS).

**Table 1.** Changes of core areas (CAs) for *Quercus acutissima* Carruth., *Q. variabilis* Blume, and *Q. chenii* Nakai in China predicted by species distributions models (SDMs) considering both climate change (CC) and land use and land cover change (LULC). Three representative concentration pathways (RCPs) were considered, including RCP 2.6, RCP 4.5, and RCP 6.0, representing different concentration trajectories from eco-friendly (RCP 2.6) to bad case (RCP 6.0).

| Years | RCPs | *Q. acutissima* | *Q. variabilis* | *Q. chenii* |
|---|---|---|---|---|
| current | | 228.66 | 217.00 | 78.33 |
| | RCP 2.6 | 221.22 | 189.32 | 48.41 |
| 2050 | RCP 4.5 | 208.98 | 180.66 | 64.86 |
| | RCP 6.0 | 215.46 | 179.98 | 51.74 |
| | RCP 2.6 | 223.45 | 198.48 | 75.41 |
| 2070 | RCP 4.5 | 207.96 | 178.80 | 68.32 |
| | RCP 6.0 | 165.42 | 158.95 | 66.71 |

For *Q. variabilis*, the suitable areas will be reduced in the future, which will shrink rapidly in southern China but expand slightly in northern China. In 2050 or 2070, the suitable habitat area was predicted to be the largest under the RCP 2.6, followed by RCP 4.5 and RCP 6.0. The CA of *Q. variabilis* will decrease in the future, and the loss of CA is the smallest under the RCP 2.6 scenario.

For *Q. chenii*, the suitable habitat area will also be reduced in the future. In 2050, it will be the largest under the RCP 6.0 scenario, followed by RCPs 4.5 and 2.6. In the RCP 2.6 scenario, the suitable habitat area was estimated to be only 66.23% of that at the present. In 2070, the suitable habitat area will increase, which will exceed the present area under the RCP 2.6 scenario, reaching $85.85 \times 10^4$ km². The CA of *Q. chenii* will be reduced in the future, and the loss of CA is the smallest under the RCP 6.0 scenario.

For all the three species, the areas of suitable habitats were predicted to be smaller under the PM scenario than under the FM scenario (Figures 4 and 5). The suitable habitat area of *Q. acutissima* under the PM scenario was shown to be more expanded in 2050. It was only reduced under the RCP 6.0 scenario of 2070, accounting for 85.84% of the current area. The future suitable habitat area of *Q. variabilis* under PM was smaller than that at the present, and the estimated habitat area in 2070 was smaller than that in 2050. The suitable habitat area of *Q. chenii* under PM will increase in all scenarios. It will increase by 34.85% under the RCP 6.0 scenario in 2050, and by 57.00% under the RCP 6.0 scenario in 2070.

## 4. Discussion

### 4.1. Key Variables Shaping Species Distributions

On the regional scale, geographical distributions of plants are mainly restricted by climatic factors, in which hydrothermal conditions play a leading role [52]. Our research found that *Q. acutissima* and *Q. variabilis* are mainly affected by AMT, AP and MTCM. One of the main factors why *Q. variabilis* and *Q. acutissima* do not expand northward under global climate change may be that their northern range limits are restricted by low temperature [27,52]. Winter temperatures in northern China are predicted to drop further in the future [53]. The extreme low-temperature events are greatly damaging to the mechanical structure of forest trees, limiting their distributions in high latitudes and altitudes [54]. At the same time, *Q. variabilis* is drought-resistant and its root tissue is more sensitive to flooding [26]. Therefore, the increase in AP and PS in southern China may explain its rapid contraction in this region [55]. However, the endemic *Q. chenii* is almost mainly affected by TS and MTCM. The influence of TS on distributions has also been found in the study of other endemic *Quercus* species [56]. TS is coupled with the seasonal variation of oak functional traits, and the narrow-distributed oak species may have worse elasticity in their functional traits [57], so it is difficult for them to adapt to more extreme seasonal changes in temperature.

Many studies showed that the contribution of land use and land cover change to SDM is not as decisive as climate factors at a large scale [58]. Our research proves this opinion and shows that climatic factors play a more important role in predicting species habitats on the macro scale. The degree of contribution of LULC to SDMs may be related to the resolution and accuracy of LULC data [17]. Under the resolution of 0.05°, human activities will not have a significant impact on the current forest cover. Although LULC is shown to have little impact on the overall range of suitable habitats, it was predicted to deeply change the landscape patterns. We found that LULC will lead to the fragmentation of suitable habitats and reduce the value of CA. Previous studies on oak trees in the western Himalayas used land use data to limit their distribution in evergreen forests [59], but evergreen forests cannot cover all the suitable habitats of *Quercus*. We also added coniferous forests as a land cover factor to predict East Asian Cerris oaks' habitats and found that the suitable areas of *Q. chenii* are more related to coniferous forests than broad-leaved forests. This may be due to the fact that they are important company tree species in coniferous forests, which always converge in the gap of pine forests [60]. Considering the huge impact of LULC on habitat fragmentation and CA, we should not ignore the effect of LULC on predicting species habitats, and we should also take the CA as a key index in future forest protection.

### 4.2. Ecological Niches of East Asian Cerris Oaks

The FM model predicts potentially colonizable areas that cannot be accessed in non-migration scenarios, which means that it predicts all ecological niches suitable for species survival [14]. The niche conservation hypothesis holds that the niche similarity between closely related species is higher than that between distant species [61]. Therefore, it is expected that species in the same group may have the same habitat change trend under CC. However, under the FM model, we found that the degree of suitable habitat change differs among the three species, although they belong to the same section of *Quercus*. For *Q. acutissima*, we found that its suitable area only changes greatly under the RCP 6.0

scenario in 2070, reflecting its robustness to climate change. For *Q. variabilis*, its habitat loss increases under more serious emission scenarios, coinciding with the general prediction that the species habitat will shrink in scenarios worse than RCP 4.5 [14,26,62]. For *Q. chenii*, its suitable habitats fluctuate greatly in different periods, which is mainly related to the high level of CC and LULC in eastern China [63]. Overall, the suitable habitats of *Q. acutissima* are predicted to be more stable, while those of *Q. chenii* are found to be more sensitive to CC and LULC.

We believe that evolutionary history is one of the main reasons why the suitable habitats of the three species have different responses to CC. *Q. chenii* was historically widely distributed in China under the warm climate in the mid-Pliocene [64]. However, the climate cooling in the Pleistocene led to the extinction of *Q. chenii* in northern China. The present populations of the species are the descendants of those who survived in the refugia of eastern China [65]. They may have adapted to the past climate but show maladaptation to future climate change. In contrast, the historical distribution of *Q. acutissima* was relatively stable. The occurrence of multiple marginal refugia contributed to the preservation of its genetic diversity under Quaternary climate fluctuations [66]. Hence, it may have stronger plasticity in response to climate change. Given that *Q. chenii* is more sensitive to CC, we suggest that it should be listed as a species requiring priority protection.

*4.3. Future Habitats under Dispersal Limitations*

Considering species dispersal limitations, the suitable areas of East Asian Cerris oaks in China are predicted to be smaller than those under the full migration scenario. An important reason for species habitat reduction under the PM scenario is that many suitable habitats will become inaccessible due to geographical isolation and habitat fragmentation [1,14]. In addition, the current actual distribution area of species is always smaller than the optimal area predicted by the SDMs, due to sampling deviation and other reasons [15]. Our results showed that the suitable habitats of *Q. chenii* and *Q. acutissima* under dispersal limitations are gradually expanding. However, the suitable habitats of *Q. variabilis* gradually decrease, which means that the new habitats of *Q. variabilis* under future CC are geographically isolated from the current habitats, while its current habitats are rapidly disappearing. Hence, we should take measures to protect the current habitats of *Q. variabilis*. Finally, the accuracy of species migration ratio is an important factor affecting habitat prediction in our study. Some studies posited that there is no migration process of oak trees in the eastern mountains under the current conditions [27], while others pointed out that with the improvement of forest management mode, the expansion of oak trees is accelerating [66]. Therefore, more evidence of seed and population dispersal may be needed to refine our study in the future. Although the partial migration rate used in this study may not accurately reflect the actual migration capacity of each species, it provides a more complete picture of potential future changes in the distributions of the studied species.

**5. Conclusions**

Our results reveal the complex effects of CC and LULC on the future distributions of East Asian Cerris oaks in China. LULC does not have a significant impact on the current range size of the three species but will lead to habitat fragmentation and a reduction in CA, which may further restrict the future migration of oaks across the landscape. Under the scenarios of CC and LULC, we found that the suitable habitats of the narrowly distributed species *Q. chenii* will be greatly reduced, while those of *Q. variabilis* in southern China will no longer be suitable for its growth. The management of current *Q. variabilis* stands should be strengthened and transplantation is needed to buffer the impacts of climate change. At the same time, protective measures should be taken to prevent the suitable habitats of *Q. chenii* from being threatened by land use and land cover change. Our work emphasizes that it is essential to take the effects of LULC and dispersal limitations into account when predicting species' habitat change in the face of climate warming.

**Supplementary Materials:** The following supporting information can be downloaded at: https://www.mdpi.com/article/10.3390/f13030367/s1, Table S1: Occurrence data of three East Asian Cerris oaks in China; Table S2: Climate variables and their abbreviations; Table S3: Pearson correlation coefficients among seven selected climate variables.

**Author Contributions:** Conceptualization, Y.C. and Y.L.; methodology, Y.C.; software, Y.C.; validation, Y.L.; resources, Y.L.; writing—original draft preparation, Y.C.; writing—review and editing, L.M.; visualization, Y.C.; supervision, L.M.; project administration, L.M.; funding acquisition, L.M. All authors have read and agreed to the published version of the manuscript.

**Funding:** This research was funded by the National Natural Science Foundation of China (31870506), the Strategic Priority Research Program of the Chinese Academy of Sciences (XDB31000000), the China Postdoctoral Science Foundation (2020M681629), and the Jiangsu Postdoctoral Research Funding Program (2021K038A).

**Institutional Review Board Statement:** Not applicable.

**Informed Consent Statement:** Not applicable.

**Data Availability Statement:** The data presented in this study are available in Supplementary Materials.

**Acknowledgments:** We thank Yuran Dong, Bingbing Xing, Jiayi Lu, Xudong Lu, Chaohe Tang, Xiuping Wu, Siyu Chen and Xiao Li from Nanjing Forestry University for their assistance in collecting species occurrence data, as well as anonymous reviewers for their helpful suggestions.

**Conflicts of Interest:** The authors declare no conflict of interest. The funders had no role in the design of the study; in the collection, analyses, or interpretation of data; in the writing of the manuscript, or in the decision to publish the results.

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
