# Peer review of "Combining the Effects of Global Warming, Land Use Change and Dispersal Limitations to Predict the Future Distributions of East Asian Cerris Oaks (Quercus Section Cerris, Fagaceae) in China"

_forests, doi:10.3390/f13030367_

Round 1
Reviewer 1 Report
This reviewer congratulates the authors for their excellent study. I think that only the title can be changed to make the study more attractive and readable by other researchers around the world.
Author Response
Response to Reviewer 1 Comments
Point 1: This reviewer congratulates the authors for their excellent study. I think that only the title can be changed to make the study more attractive and readable by other researchers around the world.
Response 1: Thank you! We agree with the reviewer’s comments.
We have changed the title of the manuscript from ‘Predicting impacts of global warming, land use change and dispersal limitations on range dynamics of East Asian Cerris oaks (Quercus section Cerris, Fagaceae) in China’ to ‘Combining the effects of global warming, land use change and dispersal limitations to predict the future distributions of East Asian Cerris oaks (Quercus section Cerris, Fagaceae) in China’. This will better illustrate the role of climate change, land use change and dispersal in species distribution models.
Reviewer 2 Report
The article presents an analysis of the future distribution of three dominant tree species of the genus Quercus, present in the subtropical deciduous forests of China. A species distribution modeling (SDM) approach is used, applying the Maxent modeling technique. The authors perform a classical SDM approach, projecting the current models under different future scenarios using bioclimatic variables. The major contribution of the work is the inclusion of land use projection, which is analyzed in combination with bioclimatic future projections. In addition, they include a method to incorporate dispersion in future bioclimatic projections.
The manuscript is considered to be very well structured and all analyses are rigorously performed and well explained. Both the graphs and maps are very well done and help to explain the analyses and results obtained. In summary, the manuscript is a contribution both methodologically and in its application for the conservation of the species analyzed.
Only a few minor points are identified, which are indicated below. Those that seek to improve the presented manuscript.
Introduction
Line 60. Reconsider the use of the acronym LUC. It is recommended to better use LULC (Land use and Land cover change) to keep the meaning of the concept addressed in the analysis. Since the land cover projection is analyzed in this specific case.
Line 69. This paragraph explains the relevance of the species selected for the analysis. It would be interesting to know if there are antecedents of the effects of climate change on the vegetation represented by the analyzed species.
Materials and methods
Line 124. Looking at the supplementary material, there are a couple of variables that have a correlation greater than 0.8. Please revise to be consistent with what is explained in the methodology.
Line 165. It would be relevant to explain in more detail, the demeter analysis, to give more clarity to the method used to change the resolution of the future land-use models.
Line 191. Justify better the use of the KISSMig technique to include dispersion. It is an old method that has not been widely used in the modeling literature. Further justification of its selection in this work would be important.
Conclusions
The conclusions are very general; better use could be made of the various results obtained. For example, the future scenarios used and the dispersion method could be discussed. For the latter, a discussion section could be added.
Author Response
Response to Reviewer 2 Comments
Point 1: Line 60. Reconsider the use of the acronym LUC. It is recommended to better use LULC (Land use and Land cover change) to keep the meaning of the concept addressed in the analysis. Since the land cover projection is analyzed in this specific case.
Response 1: Thank you! We agree with the reviewer’s comments.
We use LULC instead of LUC. LULC is more accurate than LUC and can better cover factors such as vegetation change.
Point 2: Line 69. This paragraph explains the relevance of the species selected for the analysis. It would be interesting to know if there are antecedents of the effects of climate change on the vegetation represented by the analyzed species.
Response 2: Thank you!
We supplemented other studies on the distribution changes of three East Asian Cerris oaks in China under climate change. Previous studies have shown that historical climate change has significantly affected the distributions of these oak forests. Global cooling during the Last Glacial Maximum (LGM, ~22,000 years ago) may have resulted in the southward retreat of temperate deciduous oak forests to between 22°N and 31°N. In contrast, climate warming during the mid-Holocene (~6,000 years ago) may have caused their northward expansion.
Point 3: Line 124. Looking at the supplementary material, there are a couple of variables that have a correlation greater than 0.8. Please revise to be consistent with what is explained in the methodology.
Response 3: Reply: Thank you!
The table of variable correlation in the supplementary material shows that the correlation between annual precipitation (AP) and precipitation of driest quarter (PDQ) exceeds 0.8. However, to investigate the impact of extreme precipitation on species distribution, we retained these two variables. At the same time, we added the reasons for retaining AP and PDQ in the method part of the text.
Point 4: Line 165. It would be relevant to explain in more detail, the demeter analysis, to give more clarity to the method used to change the resolution of the future land-use models.
Response 4: Reply: Thank you!
This study downloads LULC data from the open source global future land use data set, so we introduce the production and source of the global future land use data set in the method part. We have revised and improved the introduction of the global future land use dataset and added references to the demeter method in the dataset.
Point 5: Line 191. Justify better the use of the KISSMig technique to include dispersion. It is an old method that has not been widely used in the modeling literature. Further justification of its selection in this work would be important.
Response 5: Reply: Thank you!
We added a part to introduce the advantages of kISSMig model in the modification, and gave the latest example of using this model to simulate species diffusion. Species dispersal from the current range in a given ratio of migration in KISSMig. The algorithm is generally used to detect the overall migration pattern through time and provide rough estimates of the general migration rates. It is widely used in niche modeling of species with limited vagility (e.g., endangered species and plants).
Point 6: The conclusions are very general; better use could be made of the various results obtained. For example, the future scenarios used and the dispersion method could be discussed. For the latter, a discussion section could be added.
Response 6: Thank you!
We carefully revised the discussion part and split the ‘4.2 dynamics of habitat suitability’ in the manuscript into two chapters: ‘4.2 Ecological niches of East Asian Cerris oaks’ and ‘4.3 Future habitats under dispersal limitations’. In Chapter 4.3, we discuss the impact of species dispersal restrictions on the prediction of species habitats and the uncertainty of species dispersal in modelling.

Reviewer 3 Report
The authors may mention, for the different RCPs, the predicted changes in temperature and other variables for 2050 and 2070 in a table along with the assumptions considered in the BCC‐CSM1‐1 model. This will make it easier for the readers to understand the climate models.
Author Response
Response to Reviewer 3 Comments
Point 1: The authors may mention, for the different RCPs, the predicted changes in temperature and other variables for 2050 and 2070 in a table along with the assumptions considered in the BCC‐CSM1‐1 model. This will make it easier for the readers to understand the climate models.
Response 1: Thank you! We agree with the reviewer’s comments.
In order to show the differences between different RCP scenarios, we show the changes of annual mean temperature in China in the form of text in the method. In the scenario of RCP 2.6, mean annual temperature in China has increased by 1.36 °C from current to 2070, while in the RCP 4.5 scenario, it has increased by 2.13 °C and in the scenario of RCP 6.0, it has increased by 2.36 °C.